# Unusual Rapid Growth of Primary Splenic Diffuse Large B-Cell Lymphoma with Extensive Necrosis

**DOI:** 10.3390/diagnostics13010035

**Published:** 2022-12-22

**Authors:** Yue-Ren Chen, Hwa-Koon Wu, Yang-Yuan Chen

**Affiliations:** 1Division of Internal Medicine, Yuan-Lin Christian Hospital, 456 Ju-Guang Road, Yuan-Lin City 510012, Taiwan; 2Division of Radiology, Changhua Christian Hospital, 135 Nan-Xiao Street, Changhua City 50006, Taiwan; 3Division of Gastroenterology, Changhua and Yuan-Lin Christian Hospital, 135 Nanxiao Street, Changhua City 50006, Taiwan; 4Department of Hospitality Management, MingDao University, 3 Lane 138 Tai-An 2nd Street, Changhua City 50006, Taiwan

**Keywords:** primary splenic lymphoma, cystic splenic mass, doubling time

## Abstract

The primary splenic lymphoma is extremely uncommon with an incidence rate of <1% of all the lymphomas under the strict criteria for diagnosis expounded by Das Gupta et al. Clinical presentations of nonspecific symptoms are weight loss, weakness, fever, and left upper quadrant pain or discomfort due to enlarged spleen. Abdominal ultrasound and CT are the most widely used imaging modality for the assessment of lymphoma. The imaged features of splenic lymphoma are nonspecific; typical lymphoma presents as a diffusely enlarged spleen. The abdominal CT scan in our case showed a large cystic splenic mass measuring 14 cm without enhancement after contrast medium. Lymphoma is often described as an aggressive tumor because its rapid doubling time can quickly increase the size of a tumor. In our case, the tumor grew to more than 100 times its original size in 4 months. So, we present this unusual rapid growth of primary splenic lymphoma.

A 36-year-old man working as a nurse in our hospital presented with left upper quadrant (LUQ) pain, which had persisted for 1 week. He denied any past systemic disease, but the patient had undergone an abdominal ultrasound during a routine examination 4 months earlier, which had revealed a 3 cm hypoechoic splenic nodule (Figure 1). At that time, the treating physician had suggested a follow-up examination 6 months later.

Physical examination revealed a large palpable mass with local tenderness over the LUQ. The patient’s white blood cell count was 17,000/μL, with segment neutrophil 83% and lymphocyte 10%, and platelet: 475,000/mL. An abdominal computed tomographic (CT) scan revealed a large cystic splenic mass measuring 14 cm without enhancement after contrast medium (Figure 2). The mass was pushing the left kidney inward and down (Figure 3). During the initial diagnosis, a pancreatic pseudocyst involving the spleen was suspected, and the patient received an emergency splenectomy because of the rapid growth of the mass and severe pain.

The splenectomy operation removed a large solid splenic tumor. The histology exhibited a diffuse proliferation of a monotonous population of large neoplastic lymphoid cells and extensive necrosis (Figure 4). Immunohistochemical testing revealed a positive result for CD20, CD79a, BCL6, and BCL2, and a Ki-67 index greater than 90% (Figure 5). Immunohistochemical testing was also negative for CD3 and CD10. The patient was discharged without complications 1 week later. A subsequent bone marrow examination and positron emission tomography–computed tomography scan were negative. A diagnosis of primary splenic diffuse large B-cell lymphoma was recorded. The patient received CHOP regimen chemotherapy. The patient was asymptomatic upon follow-up examination and exhibited no recurrence 6 months after chemotherapy.

Lymphoma is the most common malignancy of the spleen, but primary lymphoma of the spleen is very rare with an incidence rate of <1% under the strict criteria for diagnosis expounded by Das Gupta et al. According to their criteria, diagnosis of primary splenic lymphoma should only be made when the disease is confined to the spleen but may also involve the hilar lymph nodes with no recurrence of the disease after splenectomy [1,2,3]. In our case, the patient presented with a large splenic cystic tumor that did not affect other organs, and there was no occurrence after splenectomy and chemotherapy.

Clinical presentations of nonspecific symptoms are weight loss, weakness, fever, and left upper quadrant pain or discomfort due to an enlarged spleen. CT is the most widely used imaging modality for the assessment of lymphoma. The imaged features of splenic lymphoma are nonspecific; typical lymphoma presents as a diffusely enlarged spleen. Focal lesions appear as low-density areas in the imaging with little or no enhancement following injection of intravenous contrast medium. Ahmann classified lymphomatous involvement of the spleen into the following four categories: (1) homogeneous enlargement without masses, (2) miliary masses, (3) 2–10 cm masses, and (4) a large solitary mass [4,5,6,7]. Significant necrosis may also occur with a large tumor but extensive necrosis like the necrosis observed in our case is extremely rare.

Lymphoma is often described as an aggressive tumor because its rapid doubling time can quickly increase the size of a tumor [7,8,9,10]. In our case, the tumor grew to more than 100 times its original size in 4 months, and the doubling time was only 17 days. Fortunately, the tumor in our case responded well to splenectomy with chemotherapy.

## Figures and Tables

**Figure 1 diagnostics-13-00035-f001:**
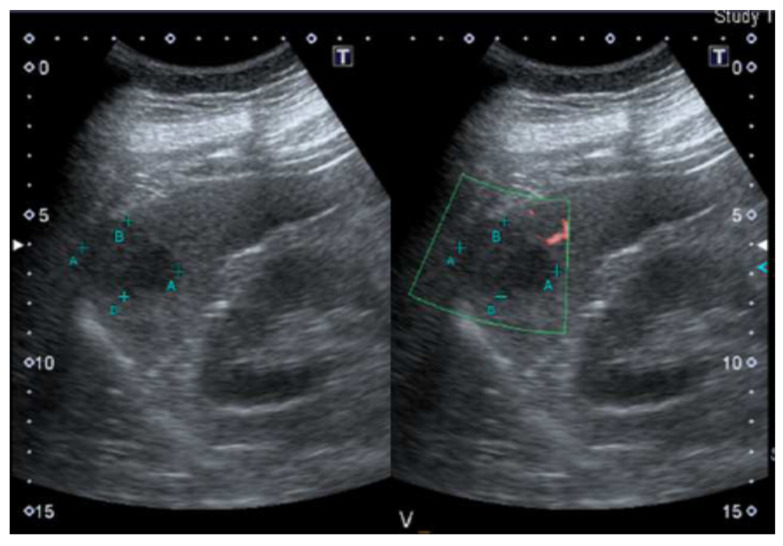
Abdominal ultrasound revealed a 3 cm hypoechoic splenic nodule during a routine examination 4 months earlier.

**Figure 2 diagnostics-13-00035-f002:**
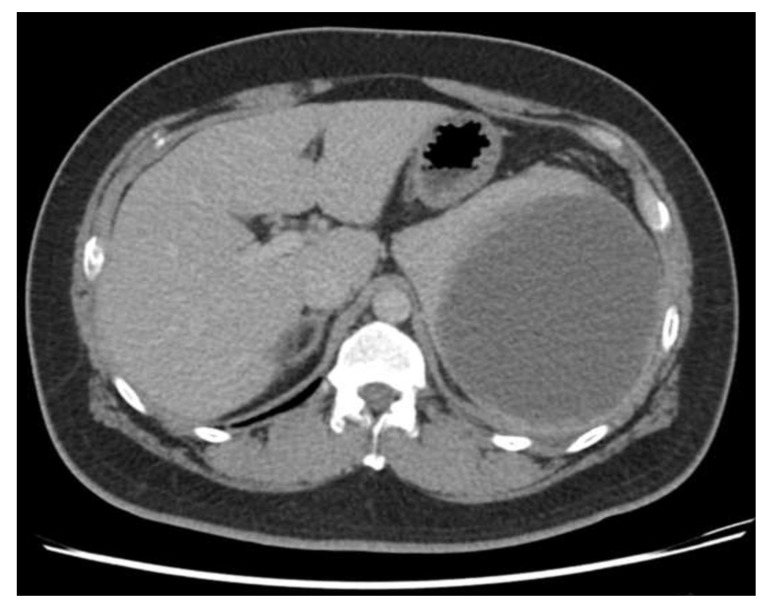
Abdominal CT scan revealed a large cystic splenic mass measuring 14 cm without enhancement.

**Figure 3 diagnostics-13-00035-f003:**
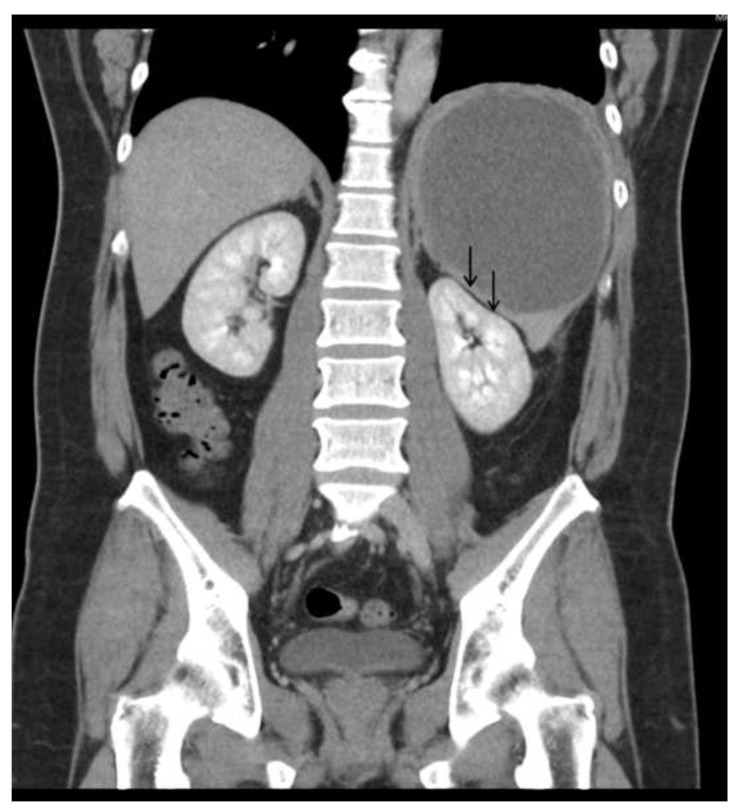
Abdominal CT scan showed the mass was pushing the left kidney inward and down.

**Figure 4 diagnostics-13-00035-f004:**
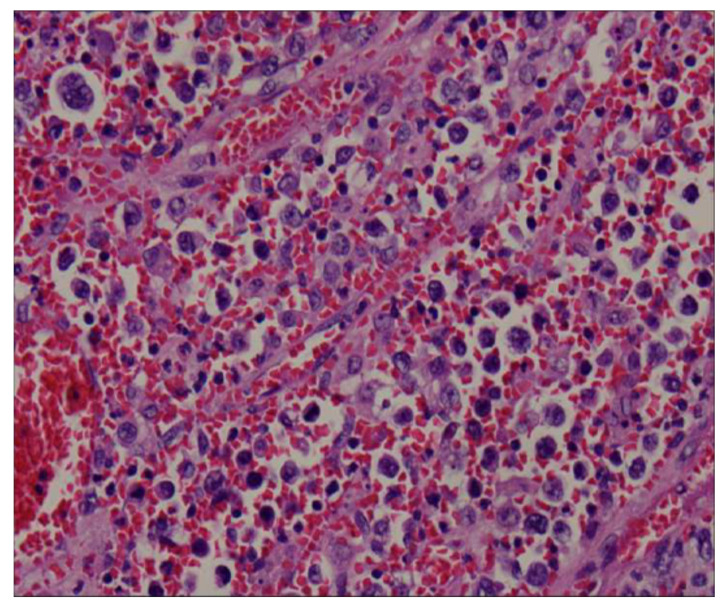
The histology revealed a diffuse proliferation of a monotonous population of large neoplastic lymphoid cells and extensive necrosis.

**Figure 5 diagnostics-13-00035-f005:**
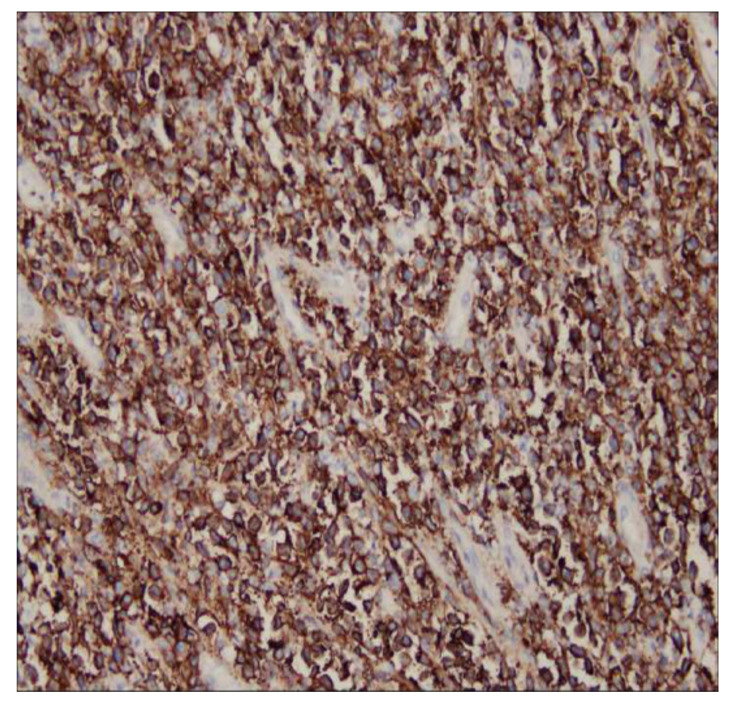
Immunohistochemical testing was positive for CD20.

## Data Availability

We did not report any data.

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
