# Peer review of "Unusual Rapid Growth of Primary Splenic Diffuse Large B-Cell Lymphoma with Extensive Necrosis"

_diagnostics, 2022, doi:10.3390/diagnostics13010035_

Round 1

Reviewer 1 Report

Dear Authors,

The subject of this case report is very interesting, but I think some comments are necessary.

The introduction is completely missing. I think a short introduction on the topic is necessary. 

I think this case report was incompletely presented. Blood count needs comments, because is highly suggestive for an infection, but nothing is mentioned in this direction. 

In this context, the discussions also need to be completed.

The references are relevant, but their position in the text does not correlate with the quoted phrases. E.g „Ahmann classified lymphomatous involve-ment of the spleen into the following 4 categories: (1) homogeneous enlargement without masses, (2) miliary masses, (3) 2–10 cm masses, and (4) a large solitary mass [4-6]”. I found Ahmann reference in the 2nd position.

I hope this comments will help you to finalize this case report.

Author Response

 Reviewer 1 report:

Point 1: The introduction is completely missing.

Response 1:

There is no introduction as rule of the image section of the diagnostics.

Point 2: Blood count needs comments

Response 2:

The patient’s white blood cell count was 17 000/μL, with segment 83% and lymphocyte 10%, and platelet: 475000/uL. The white cell count was leukocytosis with shift to left, but the clinical presentation was not likely infection.

 Point 3: The references are relevant.

 Response 3: We add the reference 11” Ahmann DL, Kiely JM, Harrison EG Jr, Payne WS. Malignant lymphoma of the spleen. A review of 49 cases in which the diagnosis was made at splenectomy. Cancer. 1966; 19(4): 461-9.”

Reviewer 2 Report

The authors report a case of splenic diffuse large B cell lymphoma with imaging.

Abstract, Line 11. “…all the lymphomas when strict criteria for diagnosis is applied. “

Delete “expounded by Das Gupta et al” unless you want to reference the paper.

Page 1, Line 24. Why did the patient undergo an abdominal ultrasound during a “routine examination” 4 months before he presented with abdominal pain?

Page 1, Line 29. “… segment 83%” should be “segmented neutrophils 83%”.

Page 5, line 62. …when the disease is confined to the spleen but may also involve the hilar lymph nodes.

Author Response

Point 1: Abstract, Line 11. “…all the lymphomas when strict criteria for diagnosis is applied. “

Delete “expounded by Das Gupta et al” unless you want to reference the pape

Response 1:

We add the reference 1.

Point 2:  Page 1, Line 24. Why did the patient undergo an abdominal ultrasound during a “routine examination” 4 months before he presented with abdominal pain?

Response 2:

This is a welfare benefit for our hospital member.

Point 3: Page 1, Line 29. “… segment 83%” should be “segmented neutrophils 83%.

Response 3:

 We change to the” segmented neutrophil 83%”.

Point 4: Page 5, line 62. …when the disease is confined to the spleen but may also involve the hilar lymph nodes.

Response 4:

We will change it.

Thank you very much for your reviewer. If you have any questions

regarding this manuscript, please do not hesitate to contact us by mail at our

correspondence address, by fax at (886)-4-7228289, by telephone at (886)-4-

7359253, or by e-mail at ychen02@gmail.com

Round 2

Reviewer 1 Report

Dear Authors,

Thank you for your explanation about white cell count. I think it could be a good point to add the explanation in the article (The white cell count indicated leukocytosis with shift to left, but the clinical presentation was not likely infection).

I have a few small corrections:

- on abstract, in he sentence Abdominal ultrasound and CT are the most widely used imaging modality for the assessment of lymphoma, imaging modality should be corrected with imaging modalities or imaging techniques.

-The splenectomy operation removed a large solid splenic tumor. I think that - Surgery removed a large solid splenic tumor - sounds better. 

- Immunohistochemical testing was positivity of CD20, CD79a..... Immunohistochemical testing showed positivity / was positive for... and Ki-67 index was greater than 90%.

- Immunohistochemical testing was also negativity of CD3 and CD10... Immunohistochemical testing also was negative for / also showed negativity

- On discussions, after Das Gupta et al, the reference should be mentioned.

- On the same paragraph - In our case, the patient presented with a large

splenic cystic tumor that did not affect other organs, and no occurrence after

splenectomy and chemotherapy/ without occurence..

- Significant necrosis may also occur with a large tumor but extensive necrosis like the necrosis observed in our case is extremely rare./ Significant necrosis may also occur whitin a large tumor... 

- Fortunately, the tumor in our case responded well to splenectomy with chemotherapy. Fortunately, in our case, the patient responded well to splenectomy with chemotherapy, without tumoral reccurence.

Author Response

Round 2:

 Reviewer 1 report:

Point 1: The white cell count indicated leukocytosis with shift to left, but the clinical presentation was not likely infection.

Response 1: We will add it in the article.

Point 2: Abdominal ultrasound and CT are the most widely used imaging modality for the assessment of lymphoma, imaging modality should be corrected with imaging modalities or imaging techniques.

Response 2: We will change “imaging modality to imaging modalities”.

Point 3: The splenectomy operation removed a large solid splenic tumor. I think that - Surgery removed a large solid splenic tumor.

Response 3: We change “operation to surgery”.

Point 4: Immunohistochemical testing was positivity of CD20, CD79a..... Immunohistochemical testing showed positivity / was positive for... and Ki-67 index was greater than 90%.

Response 4: We change “positivity of “ to “positive for” and delete “a”.

Point 5: Immunohistochemical testing was also negativity of CD3 and CD10... Immunohistochemical testing also was negative for / also showed negativity

Response 5: We change “negativity of” to ”negative for”.

- Point 6: On discussions, after Das Gupta et al, the reference should be mentioned.

Response 6: we add the reference.

- Point 7: On the same paragraph - In our case, the patient presented with a large

splenic cystic tumor that did not affect other organs, and no occurrence after

splenectomy and chemotherapy/ without occurrence.

 Response 7: We change it from no occurrence to without occurrence.

Point 8: Significant necrosis may also occur with a large tumor but extensive necrosis like the necrosis observed in our case is extremely rare./ Significant necrosis may also occur within a large tumor...

Response 8: We change it from with to within.

Point 9:  Fortunately, the tumor in our case responded well to splenectomy with chemotherapy. Fortunately, in our case, the patient responded well to splenectomy with chemotherapy, without tumor recurrence.

Response 9: We change it from “the tumor in our case responded well to splenectomy with chemotherapy” to “the patient responded well to splenectomy with chemotherapy, without tumor recurrence”.

Thank you very much for your reviewer.

Reviewer 2 Report

All queries were addressed in the revised paper, but this is not a novel case report. 

Author Response

This may be not a novel case report, but it may be an interesting case, especial rapid growth and central necrosis.